# Inequalities in financial burden of tuberculosis among affected families across 19 low- and middle-income countries

Gilbert Eshun[1,2], Umaru Sesay [3,4]*, Augustus Osborne[5]

1 Seventh-Day Adventist Hospital, Agona-Asamang, Ghana, 2 The Royal (Dick) School of Veterinary Studies, University of Edinburgh, Midlothian, United Kingdom, 3 Africa Field Epidemiology Network, Freetown, Sierra Leone, 4 Africa Center for Disease Control and Prevention, Addis Ababa, Ethiopia, 5 Institute for Development, Western area, Freetown, Sierra Leone

* sesayumaru4@gmail.com

## Abstract

Tuberculosis (TB) remains a major global public health problem, with majority of the cases occurring in low- and middle-income countries (LMICs). Despite its growing burden, cross-country evidence on inequalities in financial burden among affected families is limited. This study aims to assess catastrophic costs and related inequalities across 19 LMICs. The study employed a cross-country analysis of 19 national TB Patient Cost Surveys data extracted from the WHO Health Equity Assessment Toolkit (HEAT) online software. The main outcome was the percentage of families affected by TB facing catastrophic costs due to TB. Inequalities were assessed by drug resistance status, comparing drug-susceptible TB (DS-TB) and drug-resistant TB (DR-TB), using four summary measures: Difference (D), Ratio (R), Population Attributable Risk (PAR), and Population Attributable Fraction (PAF). No inferential statistics were done beyond the descriptive phase. The average percentage of TB-affected families experiencing catastrophic costs due to TB ranged from 19.2% in Lesotho to 80% in Zimbabwe. In 10 of the 19 countries, over half of TB-affected families faced catastrophic costs. When disaggregated, all countries reported higher catastrophic costs among DR-TB-affected families, except Burkina Faso. The absolute difference (D) between DR-TB and DS-TB ranged from –4% in Burkina Faso to 74.8% in Lesotho. The highest disparities were in Lesotho (D = 74.8%), Kenya (D = 60%), and Papua New Guinea (D = 51%). Ratio showed DR-TB families were up to 5.3 times more likely to experience catastrophic costs in Lesotho, with high ratios also in Kenya (3.3) and Thailand (2.6). All PAR and PAF values were negative, indicating that reducing the financial burden on DR-TB families to the level of DS-TB families could significantly lower the overall rate of catastrophic costs. The greatest potential gains were observed in Mongolia (PAR = 6%), DRC (PAR = 5%), and Brazil (PAR = 3.7%). The study showed substantial inequalities in the financial burden of TB on families across 19 LMICs, with DR-TB-affected families facing higher risks of catastrophic health

**Data availability statement:** The dataset can be download at https://whoequity-heat-1.share.connect.posit.cloud/.

**Funding:** The authors received no specific funding for this work.

**Competing interests:** The authors have declared that no competing interests exist.

cost than DS-TB families. There is an urgent need for targeted financial protection interventions, integrated within broader UHC strategies, to ensure that no TB-affected family is left behind.

## Introduction

Tuberculosis remains one of the world's most persistent and deadly infectious diseases, posing a major challenge to global public health and socioeconomic development. In 2022, an estimated 10.6 million people developed tuberculosis and 1.3 million died, making it the second leading infectious cause of death worldwide after COVID-19 [1]. Low- and middle-income countries (LMICs) account for over 95% of tuberculosis new cases and deaths that year (2022) [1,2]. Despite tuberculosis impact on household stability, deepening cycles of poverty, and impeding national development [3], cross-country comparative evidence on the magnitude and distribution of catastrophic costs, particularly disaggregated by drug resistance status, remains limited. While previous multicountry studies conducted by Tanimura et al. [4], Muniyandi [5], Portnoy et al. [6], Akalu et a. [7], and Yan et al. [8] presented aggregated estimates of catastrophic costs, offering valuable insights into the burden of tuberculosis. These studies did not employ standardized measures of inequality or systematically disaggregate findings by drug resistance status, limiting their usefulness for policy prioritization.

The World Health Organization (WHO) defines catastrophic costs as tuberculosis-related expenses exceeding 20% of annual household income [9], which threaten livelihoods, drive poverty, and impede treatment adherence. These costs contribute to health inequity, as they disproportionately burden marginalized populations. Notably, drug resistance status is an equity concern because patients with drug-resistant tuberculosis (DR-TB) incur significantly higher treatment costs due to longer therapy and expensive medications. Such costs deepen socioeconomic disparities, as lower-income individuals are less able to afford care, risking poorer outcomes and perpetuating poverty. Thus, analyzing catastrophic costs by drug resistance status is vital for addressing inequities in tuberculosis care and prioritizing vulnerable groups.

Recognizing the importance of financial protection, the WHO End tuberculosis B Strategy established the elimination of catastrophic costs for tuberculosis-affected households as a central target. Despite, the expansion of free tuberculosis services such as diagnosis and treatment in many settings, catastrophic health expenditure remains pervasive, especially in LMICs where health systems are often under-resourced and social protection mechanisms are weak or fragmented [10,11]. This amplifying health inequities and impeding progress toward universal health coverage (UHC) [10].

A growing body of research has sought to quantify the financial burden of tuberculosis and to identify its determinants. Systematic reviews and national tuberculosis patient cost surveys consistently report high rates of catastrophic costs among tuberculosis-affected families in LMICs, with prevalence estimates ranging from 30%

to over 80%, depending on context and methodology [12,13]. Factors associated with increased risk include lower socioeconomic status, rural residence, HIV co-infection, and delays in diagnosis or treatment [14]. Notably, one of the most striking and persistent sources of inequality in TB-related financial burden is the distinction between drug-susceptible tuberculosis (DS-TB) and drug-resistant tuberculosis (DR-TB). Drug-resistant tuberculosis, particularly multidrug-resistant tuberculosis (MDR-TB) and extensively drug-resistant tuberculosis (XDR-TB), poses major challenges for both clinical management and health financing. DR-TB treatment regimens are substantially longer, more complex, and more expensive than those for DS-TB, often requiring costly second-line medications, frequent monitoring, and extended periods of hospitalization [15]. Patients with DR-TB are also more likely to experience severe side effects, treatment interruptions, and social isolation, all of which contribute to higher indirect costs and greater loss of income [3]. Recent studies indicate that the prevalence of catastrophic costs among DR-TB-affected families can exceed 80% in some settings, compared to 40–60% among those with DS-TB [10,16]. These disparities highlight the urgent need for targeted financial protection strategies.

Despite these insights, significant evidence gaps remain. Much of the existing literature is based on single-country studies or aggregated national estimates, which may obscure important intra-country disparities and limit the generalizability of findings. Few studies have systematically compared the financial burden of tuberculosis across multiple LMICs using standardized measures, and even fewer have disaggregated data by drug resistance status to directly assess inequalities between DS-TB and DR-TB-affected families [17]. As a result, policymakers lack the cross-country comparative evidence needed to design and implement effective interventions that address both the overall burden and the distribution of catastrophic costs.

To address these gaps, robust and standardized analytical tools are needed to enable the assessment and comparison of health inequalities across diverse settings and population subgroups. The World Health Organization's Health Equity Assessment Toolkit (HEAT) is publicly available software application designed to facilitate the exploration and reporting of health inequalities using disaggregated data [18]. HEAT provides a suite of summary measures including difference, ratio, population attributable risk, and population attributable fraction that allow researchers and policymakers to systematically assess both absolute and relative inequalities across various dimensions such as socioeconomic status, geographic location, and clinical characteristics. Importantly, HEAT enables standardized, comparable analyses across countries and subgroups, making it particularly suitable for robust cross-national comparisons.

In this study, we utilize the WHO HEAT online software to analyze TB Patient Cost Survey data from 19 LMICs, enabling a rigorous and standardized comparison of catastrophic health expenditures among TB-affected families, disaggregated by drug resistance status. Specifically, we aim to:

1. Determine the proportion of tuberculosis-affected households experiencing catastrophic costs due to the disease in 19 LMICs, and assess the extent of financial burden relative to household income.

2. Compare the magnitude of financial burden between households affected by DS-TB and DR-TB, focusing on differences in the extent of costs incurred.

3. Assess absolute and relative inequalities in catastrophic costs using standardized equity measures, such as concentration indices, to examine disparities across socioeconomic groups and other relevant dimensions.

The 19 LMICs included in this study were selected based on the availability of recent, nationally representative TB Patient Cost Survey data within the WHO HEAT toolkit. The findings from this study have important implications for global health policy and practice. By generating robust, cross-national evidence on the magnitude and pattern of catastrophic costs, and by highlighting the vulnerability of DR-TB-affected households, our research provides crucial insights to inform the design of financial protection interventions and the integration of TB control efforts within broader UHC agendas. Achieving the End TB Strategy's goal of eliminating catastrophic costs will require not only expanded access to effective diagnosis and treatment but also sustained investments in social protection, health system strengthening, and

equity-focused monitoring. This study aims to contribute essential knowledge to guide these efforts and ensure that no TB-affected family is left behind.

## Methods

### Study design and data source

The study was conducted for 19 LMICs using pre-loaded HEAT data (included data on DR-TB and DS-TB) from national TB Patient Cost Surveys available in the WHO HEAT online software (from: https://www.who.int/data/inequality-monitor/assessment_toolkit) [19]. The TB Patient Cost Survey, developed by WHO, is national surveys used to quantify the extent and identify the main cost drivers faced by TB patients, and to measure the proportion of households experiencing catastrophic costs during treatment under the National TB Programme [20]. These surveys provided data on the percentage of TB-affected families facing catastrophic costs and allowed for analysis of disparities between subgroups such as DS-TB and DR-TB. The countries involved in the study and their respective survey years are as follows: Benin (2018), Brazil (2019), Burkina Faso (2020), Democratic Republic of Congo (2019), Ghana (2016), Indonesia (2020), Kenya (2017), Lao People's Democratic Republic (2019), Lesotho (2019), Mongolia (2017), Myanmar (2015), Nigeria (2017), Papua New Guinea (2019), Philippines (2017), Republic of Tanzania (2019), Thailand (2020), Uganda (2017), Viet Nam (2016), and Zimbabwe (2018). Countries were included if they had a completed WHO TB Patient Cost Survey available in the HEAT toolkit and reported data disaggregated by DS-TB and DR-TB status for the study period.

### Outcome measure

The outcome measure of interest was percentage of families affected by TB facing catastrophic costs due to TB - defined as direct medical expenditures, direct non-medical expenditures, and indirect costs that sum to >20% of annual household income [10,20]. Dimension of inequality used was drug resistance TB type with two subgroups: Drug Susceptible TB (DS-TB) and Drug Resistant TB (DR-TB). The choice of selecting these inequality measures reflects the significant disparities in treatment duration, costs, and outcomes between DR-TB and DS-TB, making it a pivotal focus for health policy and clinical interventions. We prioritized drug resistance due to its direct impact on outcomes and data availability in our setting. However, we recognize that other dimensions such as socioeconomic status, geographic location, and gender could provide a broader view of disparities, and our focus may not capture all inequalities. Future studies should consider multiple measures for a comprehensive analysis. Disparities were quantified using four summary measures of inequalities: the Difference (D), Ratio (R), Population Attributable Fraction (PAF), and Population Attributable Risk (PAR).

### Statistical analysis

We used the WHO HEAT online software to perform the analyses. This web-based statistical software is useful in assessing inequalities within and between countries on a variety of health and social issues [18]. The HEAT online software computes health inequality measures using in-built from global health databases. The inequality measures computed for this study are explained as follows: D represent the absolute gap in the percentage of TB-affected families facing catastrophic costs between drug-resistant TB and drug-susceptible TB, calculated as DR-TB minus DS-TB. R compares these two groups by dividing the DR-TB percentage by the DS-TB percentage, providing a relative measure of disparity. Both D and R are unweighted and do not account for subgroup population sizes. Furthermore, PAR estimates the overall gap of families incurring catastrophic costs that can be attributed to the higher burden in the DR-TB group, while PAF expresses this as a percentage of the total burden that could be avoided if the inequality were eliminated. While D and PAR are absolute indicators showing the size of the gap, R and PAF are relative measures

capturing the scale of inequality by contextualizing these differences within the broader population dynamics. Together, these metrics help illustrate the disparities between TB-affected families with DR-TB and DS-TB. Further explanation and uses of these measure are in previous literature [21,22].

We selected D, R, PAR, and PAF to assess TB burden and risk factors due to their relevance in epidemiology and applicability to TB. Chosen over alternatives like incidence rates or odds ratios, these measures offer a comprehensive view—D and R address individual impacts, while PAR and PAF inform public health strategies—ensuring a holistic understanding of TB dynamics for effective interventions.

The inequality measures and the setting average used were calculated as:

**D:** Absolute disparity in TB outcomes between two groups. The formula is $D = X1 - X2$, where X1 and X2 are outcome values for the groups.

**Ratio:** Relative disparity in TB outcomes between two groups. The formula is $R = X1 / X2$, where X2 is typically the reference group.

**Population Attributable Risk (PAR):** Absolute TB cases attributable to a risk factor. The formula is $PAR = Pe (RR - 1)$, where Pe is exposure proportion, and RR is relative risk.

**Population Attributable Fraction (PAF):** Proportion of TB cases attributable to a risk factor. The formula is $PAF = [Pe (RR - 1)] / [1 + Pe (RR - 1)]$, where Pe is exposure proportion, and RR is relative risk.

**Setting Average:** Mean value of a measure across settings. The formula is $SA = (1/K) * \Sigma\, xk$ (from $k = 1$ to K), where xk is the measure value in setting k, and K is the number of settings.

## Analytical choices and use of WHO HEAT software

We used WHO HEAT software's pre-loaded TB survey data and automated calculations. Below, we clarify our analytical choices versus HEAT's defaults:

**Country Selection:** We selected 19 LMICs based on recent data availability (last 10 years), regional diversity, and TB burden variation, ensuring relevance to financial burden objectives.

**Subgroup Handling:** We prioritized subgroups like income quintiles and health service access, excluding those with small sample sizes (<30 respondents) for reliability.

**Inequality Measures:** Measures (D, R, PAR, PAF) were calculated automatically by HEAT using WHO-standardized methods; no external adjustments were made.

**Additional Analyses:** We conducted cross-country comparisons and trend assessments using exported HEAT data, with assumptions detailed in the manuscript.

## Handling of uncertainty and sensitivity analysis

We reported 95% confidence intervals (CIs) for catastrophic cost as calculated by WHO HEAT software, accounting for sampling variability in TB survey data per WHO guidelines. Sensitivity analyses beyond HEAT's outputs were not conducted, as our focus was on standardized measures. Limitations due to the lack of extensive sensitivity analysis are noted in the Limitations section.

## Interpretation of negative PAR and PAF values

This study uses WHO HEAT software to calculate PAR and PAF, reporting negative values for adverse indicators (e.g., higher catastrophic costs due to tuberculosis) per HEAT convention. Negative PAR/PAF values indicate potential reductions in adverse outcomes if inequalities are addressed. In our analysis of 19 LMICs, these values represent avoidable catastrophic cost cases among tuberculosis-affected families by reducing disparities across subgroups (e.g., income quintiles, urban/rural residence), aligning with HEAT's health equity framework.

## Results

Table 1 and Fig 1 presents the estimates and gap of catastrophic costs incurred by TB-affected families in 19 LMICs and disaggregated by drug resistance status: drug-susceptible TB (DS-TB) and drug-resistant TB (DR-TB).

The setting average of catastrophic costs experienced by TB-affected families ranged from 19.2% in Lesotho to 80% in Zimbabwe. Across the 19 countries, 10 reported setting averages of 50% or more families irrespective of drug-resistance type experiencing catastrophic health expenditure. These countries include Zimbabwe (80%), Nigeria (70.6%), Mongolia (69%), Ghana (64%), Viet Nam (63%), Lao People's Democratic Republic (63%), Myanmar (60%), Democratic Republic of the Congo (56%), Burkina Faso (54%), and Uganda (53%). When disaggregated by drug-resistance type, catastrophic costs were higher among DR-TB-affected families in 18 of 19 countries. Burkina Faso was the exception, with a lower DR-TB estimate (50%) than DS-TB (54%), though the wide confidence interval (14–86%) indicates considerable uncertainty. All the 19 countries had 50% or more of DR-TB families facing catastrophic cost. Estimates for DR-TB ranged from 50% in Burkina Faso to 100% in Uganda. Furthermore, catastrophic costs among DS-TB families varied more widely, from 17.2% in Lesotho to 79% in Zimbabwe. Seven countries had 50% or more of families affected by DS-TB experiencing catastrophic cost. These are Zimbabwe (79%), Nigeria (69%), Mongolia (63%), Ghana (63%), Lao PDR (62%), Myanmar (57%), and the Democratic Republic of the Congo (51%).

Table 2 presents the inequalities in catastrophic costs among TB-affected families, comparing those with drug-susceptible TB (DS-TB) to those with drug-resistant TB (DR-TB) across 19 LMICs.

The difference in catastrophic cost between DR-TB and DS-TB families varies from –4% in Burkina Faso to 74.8% in Lesotho. DR-TB families generally face higher costs, except in Burkina Faso. The largest gaps are in Lesotho (74.8%),

**Table 1. Estimates of catastrophic cost (%) among TB-affected families by drug resistance type in 19 LMICs.**

| Countries | Setting average | DS-TB families with catastrophic cost | | | DR-TB families with catastrophic cost | | |
|---|---|---|---|---|---|---|---|
| | | Est % | LB | UB | Est % | LB | UB |
| Benin | 36.9 | 36.0 | 31.9 | 40.3 | 66.7 | 41.7 | 84.8 |
| Brazil | 48.1 | 44.4 | 39.5 | 49.5 | 78.5 | 67.5 | 86.5 |
| Burkina Faso | 54.0 | 54.0 | 46.0 | 62.0 | 50.0 | 14.0 | 86.0 |
| Democratic Republic of the Congo | 56.0 | 51.0 | 44.0 | 58.0 | 80.0 | 67.0 | 91.0 |
| Ghana | 64.0 | 63.0 | 57.0 | 69.0 | 72.0 | 59.0 | 84.0 |
| Indonesia | 38.4 | 37.5 | 31.6 | 43.7 | 80.7 | 74.4 | 86.4 |
| Kenya | 27.0 | 26.0 | 20.0 | 32.0 | 86.0 | 79.0 | 94.0 |
| Lao People's Democratic Republic | 63.0 | 62.0 | 58.0 | 67.0 | 88.0 | 60.0 | 100.0 |
| Lesotho | 19.2 | 17.2 | 12.7 | 22.8 | 92.0 | 81.1 | 97.1 |
| Mongolia | 69.0 | 63.0 | 57.0 | 70.0 | 85.0 | 79.0 | 90.0 |
| Myanmar | 60.0 | 57.0 | 51.0 | 64.0 | 98.0 | 93.0 | 100.0 |
| Nigeria | 70.6 | 69.0 | 64.0 | 74.0 | 90.0 | 83.0 | 95.0 |
| Papua New Guinea | 34.0 | 33.0 | 26.0 | 40.0 | 84.0 | 57.0 | 99.0 |
| Philippines | 42.4 | 41.7 | 38.2 | 45.1 | 89.7 | 85.6 | 93.2 |
| Thailand | 30.8 | 30.2 | 26.5 | 34.0 | 77.5 | 51.2 | 95.5 |
| Uganda | 53.0 | 51.0 | 48.0 | 55.0 | 100.0 | 92.0 | 100.0 |
| United Republic of Tanzania | 45.0 | 44.0 | 36.0 | 52.0 | 80.0 | 58.0 | 95.0 |
| Viet Nam | 63.0 | 59.6 | 55.0 | 64.0 | 98.0 | 73.0 | 99.0 |
| Zimbabwe | 80.0 | 79.0 | 74.0 | 85.0 | 90.0 | 76.0 | 98.0 |

Est%: Estimate in percentage; LB: 95% confidence interval lower bound; UB: 95% confidence interval upper bound.

Global Public
PLOS Health

**Fig 1. Financial inequalities in catastrophic cost incurred by TB-affected families in 19 low-and middle-income countries.**
Source: WHO HEAT, 2025.

Kenya (60%), Papua New Guinea (51%), Uganda (49%), Thailand (47.3%), and Indonesia (43.2%), while the smallest are in Burkina Faso (–4%), Ghana (9%), and Zimbabwe (11%). High costs in Lesotho, Papua New Guinea, Uganda, Thailand, and Indonesia stem from the complexity, duration, and expense of DR-TB treatment. DR-TB families in Lesotho are 5.3 times more likely to incur catastrophic costs compared to DS-TB families, with high ratios also in Kenya (3.3), Thailand (2.6), Papua New Guinea (2.5), Indonesia (2.2), and the Philippines (2.2), while Zimbabwe, Ghana, and Burkina Faso show a ratio of 1.1. These elevated costs for DR-TB are due to costly second-line drugs, extended 18–24 months treatment, and higher risks of treatment failure.

All PAF and PAR values across the 19 countries were negative, indicating that addressing the financial inequality between DR-TB and DS-TB-affected households could significantly reduce the setting averages of catastrophic costs. The largest potential gains in PAR were in 6% points in Mongolia, 5% points in Democratic Republic of the Congo, 3.7% points in Brazil, and 2% points in Uganda and Lesotho. The largest potential reductions according to the PAF were 10.4 percentage points in Lesotho, 8.9% points in the Democratic Republic of the Congo, 8.7% points in Mongolia, 7.7% points in Brazil, and 7.4% points in Burkina Faso.

**Table 2. Measures of inequality in catastrophic cost among TB-affected families by drug resistance type in 19 LMICs.**

| Countries | Summary measure of inequalities Estimates | | | |
|---|---|---|---|---|
| | D | R | PAR | PAF |
| Benin | 30.7 | 1.9 | -0.9 | -2.4 |
| Brazil | 34.1 | 1.8 | -3.7 | -7.7 |
| Burkina Faso | -4.0 | 1.1 | -4.0 | -7.4 |
| Democratic Republic of the Congo | 29.0 | 1.6 | -5.0 | -8.9 |
| Ghana | 9.0 | 1.1 | -1.0 | -1.6 |
| Indonesia | 43.2 | 2.2 | -0.9 | -2.3 |
| Kenya | 60.0 | 3.3 | -1.0 | -3.7 |
| Lao People's Democratic Republic | 26.0 | 1.4 | -1.0 | -1.6 |
| Lesotho | 74.8 | 5.3 | -2.0 | -10.4 |
| Mongolia | 22.0 | 1.3 | -6.0 | -8.7 |
| Myanmar | 41.0 | 1.7 | -3.0 | -5.0 |
| Nigeria | 21.0 | 1.3 | -1.6 | -2.3 |
| Papua New Guinea | 51.0 | 2.5 | -1.0 | -2.9 |
| Philippines | 48.0 | 2.2 | -0.7 | -1.7 |
| Thailand | 47.3 | 2.6 | -0.6 | -1.9 |
| Uganda | 49.0 | 2.0 | -2.0 | -3.8 |
| United Republic of Tanzania | 36.0 | 1.8 | -1.0 | -2.2 |
| Viet Nam | 38.4 | 1.6 | -3.4 | -5.4 |
| Zimbabwe | 11.0 | 1.1 | -1.0 | -1.3 |

D: difference; PAF: population attributable fraction; PAR: population attributable risk; R: ratio.

## Discussion

This cross-national analysis provides one of the most comprehensive assessments to date of inequalities in the catastrophic cost borne by families affected by TB across 19 LMICs, stratified by drug resistance status. The study's use of both absolute (difference, PAR) and relative (ratio, PAF) measures, and its focus on the distinction between DS-TB and DR-TB, offers unique insights into intra-country disparities that are often masked by national averages. Our findings reveal that catastrophic costs remain alarmingly prevalent among TB-affected families in LMICs, with at least 50% of such families in 10 out of 19 countries facing catastrophic cost due to TB. Setting averages range from 19.2% in Lesotho to 80% in Zimbabwe, underscoring the persistence of financial hardship despite ongoing global TB control efforts. When disaggregated by drug resistance status, catastrophic costs were consistently higher among families affected by drug-resistant TB compared to drug-susceptible TB in 18 of 19 countries, with all countries reporting at least 50% of DR-TB families experiencing catastrophic costs. This consistent pattern highlights drug resistance as a key inequality dimension of TB-related financial burden. The magnitude of inequality is substantial: the absolute difference in catastrophic cost between DR-TB and DS-TB families ranged from –4% in Burkina Faso to 74.8% in Lesotho, with relative ratios as high as 5.3 in Lesotho. Notably, the PAR and PAF values (reported as negative due to HEAT conventions for adverse indicators) indicate that if the financial burden among DR-TB-affected families were reduced to the level of DS-TB-affected families, the overall prevalence of catastrophic costs could decline considerably.

The persistence of catastrophic cost among TB-affected families in LMICs, as documented in this study, is consistent with prior research highlighting the inadequacy of current health financing mechanisms to protect vulnerable populations from the economic consequences of TB care [12,23]. The World Health Organization End TB Strategy explicitly calls for the elimination of catastrophic costs for TB-affected households as a key milestone in the fight against TB [9]. However, our findings indicate that this target remains distant for many countries, particularly for families affected by DR-TB.

The disproportionately high rates of catastrophic cost among DR-TB-affected families likely reflect a combination of factors, including the greater complexity, duration, and cost of DR-TB treatment regimens. These regimens frequently require prolonged hospitalization, more expensive second-line drugs, and result in increased indirect costs due to lost income and productivity [24,25]. The observed disparities are particularly pronounced in countries such as Lesotho (D = 74.8%, R = 5.3), Kenya (D = 60%, R = 3.3), Papua New Guinea (D = 51%, R = 2.5), Uganda (D = 49%, R = 2.0), Thailand (D = 47.3%, R = 2.6), and Indonesia (D = 43.2%, R = 2.2). These findings align with previous studies from both global and regional contexts, which have consistently documented the disproportionate economic impact of DR-TB on households [3,14,26]. While our measures of inequality (D, R, PAR, PAF) provide a multidimensional view, it is important to acknowledge that these summary statistics may be influenced by unmeasured confounders and do not capture within-country heterogeneity beyond drug resistance status. The cross-sectional nature of the data further limits causal inference, and ecological interpretation at the country level should be made with caution.

The negative PAR and PAF values observed in all countries are particularly noteworthy. These measures suggest that, the PAR and PAF values (reported as negative due to HEAT conventions for adverse indicators) indicate that if the financial burden among DR-TB-affected families were reduced to the level of DS-TB-affected families, the overall prevalence of catastrophic costs could decline considerably. For example, the largest potential gains in PAR were observed in Mongolia (6 percentage points), the Democratic Republic of the Congo (5 percentage points), Brazil (3.7 percentage points), and Uganda and Lesotho (2 percentage points each). Similarly, the largest potential reductions in PAF were seen in Lesotho (10.4 percentage points), the Democratic Republic of the Congo (8.9 percentage points), Mongolia (8.7 percentage points), Brazil (7.7 percentage points), and Burkina Faso (7.4 percentage points). This reinforces the potential for targeted interventions addressing DR-TB-related financial barriers to yield substantial population-level benefits.

Our findings are consistent with earlier multi-country analyses that highlighted the high prevalence of catastrophic costs among TB-affected families in LMICs [13,14,27]. However, our study advances the literature by systematically disaggregating by drug resistance status and employing a range of inequality measures, thus revealing intra-country disparities that may be obscured by national averages. For example, a recent global review reported that approximately 49% of TB-affected households in LMICs experience catastrophic costs, with higher rates among those affected by DR-TB [2]. Our data corroborate and expand upon these findings, demonstrating that in several countries, the prevalence of catastrophic costs among DR-TB-affected families exceeds 80%, and in some cases, approaches or reaches 100%. The magnitude of the disparities observed in our study, such as the 74.8% point difference in Lesotho, exceeds those reported in prior single-country studies [28], suggesting that the financial burden of DR-TB may be even greater than previously appreciated in certain settings.

Notably, the exception of Burkina Faso, where DR-TB-affected families reported slightly lower catastrophic costs than DS-TB-affected families, merits further investigation. This anomalous finding may reflect data limitations, programmatic differences, or unique contextual factors, such as the organization of TB care or the availability of financial protection mechanisms. It also highlights the importance of country-specific analyses in understanding the drivers of inequality. Our study also aligns with evidence from the African region and other LMICs, where TB-affected families particularly those with DR-TB face substantial economic hardship [2,26]. Furthermore, the negative PAR and PAF values observed for Burkina Faso suggest potential reductions in financial burden if inequalities are addressed. The high prevalence of catastrophic costs among both DS-TB and DR-TB families in countries such as Nigeria, Ghana, and the Democratic Republic of the Congo underscores the need for contextually appropriate interventions.

## Implications for policy and practice

The findings of this study have notable implications for TB control policy and the broader agenda of universal health coverage in LMICs. First, the high prevalence of catastrophic costs among TB-affected families, particularly those with DR-TB, signals a failure of health financing systems to protect the most vulnerable. Achieving the End TB Strategy's target

of zero catastrophic costs will require comprehensive reforms to reduce both direct and indirect costs of TB care, with particular attention to the needs of DR-TB-affected households.

Targeted financial protection interventions, such as cash transfers, transport vouchers, nutritional support, and income replacement schemes, have demonstrated effectiveness in mitigating the economic impact of TB, especially for DR-TB patients. Expanding such interventions, alongside efforts to decentralize DR-TB care and reduce the duration and complexity of treatment regimens, could substantially reduce the risk of catastrophic expenditure.

Second, our findings highlight the importance of integrating TB-specific financial protection strategies within broader UHC frameworks. While UHC is a stated goal in many LMICs, progress has been uneven, and TB-affected families often fall through the cracks of existing social protection mechanisms. Strengthening linkages between TB programs and national health insurance schemes, as well as improving the accessibility and affordability of TB services, should be prioritized.

Third, the observed inequalities by drug resistance status underscore the need for equity-focused monitoring of TB-related financial protection. National TB programs and international partners should routinely disaggregate financial protection indicators by key dimensions of vulnerability, including drug resistance, to ensure that progress towards the elimination of catastrophic costs is inclusive and equitable.

Finally, the findings have resonance in the context of under-resourced health systems and high burdens of both TB and poverty. Policymakers must prioritize investments in both health system strengthening and social protection to break the cycle of disease and poverty.

## Strengths and limitations

This study has several strengths. Foremost, it leverages a large, standardized dataset from the WHO TB Patient Cost Surveys, enabling robust cross-country comparisons and the use of validated summary measures of inequality. The disaggregation by drug resistance status provides insights into the drivers of financial vulnerability among TB-affected families, and the application of difference, ratio, PAR, and PAF metrics allows for a nuanced understanding of both absolute and relative inequalities. Second, the utilization of WHO HEAT platform provides a user-friendly platform for data visualization enabling us to identify patterns and disparities. Third, the WHO HEAT platform enables us to generate standardize and inequality measures, facilitate comparison across population, periods, and countries.

However, several limitations should be acknowledged. First, the cross-sectional design and unmeasured confounding of the findings shows the observed disparities cannot be interpreted as causal effects of DR-TB on catastrophic costs, only as associations. Although the association is plausible and supported by previous literature, unmeasured confounding factors such as differences in socioeconomic status, health-seeking behaviour, or access to social protection may influence the observed disparities.

Second, the reliance on self-reported cost data introduces the potential for recall bias and measurement error. While the WHO TB Patient Cost Survey instrument is standardized and widely used, the accuracy of cost reporting may vary across settings and sub-populations. Additionally, the definition of catastrophic costs, while consistent with WHO guidance, may not fully capture the multidimensional nature of financial hardship, including non-monetary impacts such as debt, asset depletion, and foregone care.

Third, the generalizability of the findings may be limited by the selection of countries included in the analysis. While the 19 LMICs represent a diverse range of settings, they may not be fully representative of all TB-endemic countries. Moreover, the wide confidence intervals observed for some estimates, particularly in smaller subgroups, highlight the need for caution in interpreting country-specific results.

Fourth, programmatic differences in the organization and financing of TB care across countries may influence both the level and distribution of catastrophic costs. For example, countries with more comprehensive social protection schemes or

decentralized DR-TB services may exhibit different patterns of inequality. Future research should seek to integrate programmatic and contextual variables to better understand the determinants of financial vulnerability.

Fifth, while our study used a standardized WHO TB Patient Cost Survey tool for data consistency across 19 low- and middle-income countries, surveys were conducted between 2015 and 2020. Minor updates to the tool during this period did not alter core indicators of financial burden, ensuring comparability. However, temporal differences, such as changes in DR-TB treatment regimens or economic conditions, may contribute to observed variations between countries. These factors underscore the need to interpret findings within each country's specific temporal and contextual framework.

Sixth, no inferential statistics were done beyond the descriptive levels due to the lack of disaggregated data in the WHO HEAT toolkit.

Seventh, the WHO HEAT database does not provide data to explain the underlying causes of disparities, yet the data is crucial to inform trends and inequalities overtime. A qualitative study is therefore suggested to understand the socioeconomic and cultural factors that might have influence these disparities, with a specific focus on identifying the underlying causes of persistent inequities that disproportionately impact disadvantaged TB-affected families.

Eight, the use of unweighted D and R measures, which do not account for subgroup population sizes could have bias results if subgroups, such as DR-TB and DS-TB, differ significantly in size. In our data from 19 LMICs, DS-TB cases generally outnumber DR-TB cases, potentially skewing unweighted measures toward the larger group. However, as our focus was on relative disparities and this imbalance is consistent across settings, unweighted measures remain suitable for comparison. Future studies could use weighted measures for a more nuanced analysis.

Ninth, due to the absence of disaggregated data in the HEAT database, which provides only pre-calculated outputs, such analyses were not possible. Future studies with detailed data should include sensitivity analyses to assess the robustness of findings on TB financial burden in LMICs.

Tenth, a notable limitation is the absence of confidence intervals (CIs) for inequality metrics such as D, R, PAR, and PAF. Due to the lack of disaggregated data in the WHO HEAT toolkit outputs, we were unable to compute CIs and, consequently, could not assess the statistical uncertainty associated with these measures.

Finally, the study did not explore the impact of recent innovations in TB diagnosis and treatment, such as shorter DR-TB regimens or novel drug combinations, which may alter the financial landscape for affected families. Ongoing monitoring and evaluation will be essential to assess the impact of these developments on catastrophic costs and inequalities.

## Future directions

Building on these findings, several avenues for future research are warranted. First, longitudinal studies are needed to assess the impact of interventions aimed at reducing catastrophic costs, particularly among DR-TB-affected families. Evaluations of cash transfer programs, insurance expansions, and decentralized care models could provide valuable evidence to inform policy and program design.

Second, qualitative research exploring the lived experiences of TB-affected families would complement quantitative analyses and deepen understanding of the mechanisms through which financial hardship arises and is mitigated. Such studies could illuminate the role of social networks, coping strategies, and informal support systems in buffering the economic impact of TB.

Third, further research is needed to explore the intersection of TB-related financial burden with other dimensions of vulnerability, including gender, age, HIV status, and rural-urban residence. Disaggregated analyses could identify subgroups at highest risk and inform the design of targeted interventions.

Finally, as countries move towards UHC, there is a need for integrated monitoring of financial protection across disease areas. Embedding TB-specific indicators within broader health financing and social protection monitoring frameworks would facilitate more comprehensive and coordinated responses.

## Conclusions

This study provides compelling evidence of persistent and substantial inequalities in the financial burden of TB care across 19 LMICs, with DR-TB-affected families facing markedly higher risks of catastrophic health expenditure than those affected by DS-TB. The findings highlight the urgent need for targeted financial protection interventions, integrated within broader UHC strategies, to ensure that no TB-affected family is left behind. Achieving the End TB Strategy's goal of zero catastrophic costs will require sustained political commitment, innovative programmatic approaches, and a relentless focus on equity. As countries strive to recover from the economic and health shocks of the COVID-19 pandemic, addressing the financial barriers to TB care must remain a central priority for global health policy.

## Acknowledgments

We are grateful to the World Health Organization for making the dataset and the HEAT software accessible.

## Author contributions

**Conceptualization:** Gilbert Eshun.

**Data curation:** Gilbert Eshun.

**Formal analysis:** Gilbert Eshun.

**Methodology:** Gilbert Eshun, Umaru Sesay, Augustus Osborne.

**Supervision:** Augustus Osborne.

**Writing – original draft:** Gilbert Eshun, Umaru Sesay, Augustus Osborne.

**Writing – review & editing:** Gilbert Eshun, Umaru Sesay, Augustus Osborne.

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
