## [Decision Letter · Decision Letter 0]

11 Jan 2026

PGPH-D-25-02325

INEQUALITIES IN THE FINANCIAL BURDEN OF TUBERCULOSIS AMONG AFFECTED FAMILIES: EVIDENCE FROM 19 LOW- AND MIDDLE-INCOME COUNTRIES

Dear Dr. Sesay,

Thank you for submitting your manuscript to PLOS Global Public Health. After careful consideration, we feel that it has merit but does not fully meet PLOS Global Public Health’s publication criteria as it currently stands. Therefore, we invite you to submit a revised version of the manuscript that addresses the points raised during the review process.

We look forward to receiving your revised manuscript.

Kind regards,

Alice Zwerling, PhD

Academic Editor

Journal Requirements:

1. Please amend your online Financial Disclosure statement. If you did not receive any funding for this study, please simply state: “The authors received no specific funding for this work.”

2. Please update your online Competing Interests statement. If you have no competing interests to declare, please state: “The authors have declared that no competing interests exist.”

3. Please provide separate main figure files in .tif or .eps format only and remove any figures embedded in your manuscript file. Please also ensure that all files are under our size limit of 10MB. Please leave the figure captions in the manuscript.

Additional Editor Comments (if provided):

Reviewers' comments:

Reviewer's Responses to Questions

**Comments to the Author**

1. Does this manuscript meet PLOS Global Public Health’s publication criteria ? Is the manuscript technically sound, and do the data support the conclusions? The manuscript must describe methodologically and ethically rigorous research with conclusions that are appropriately drawn based on the data presented.

Reviewer #1: Yes

Reviewer #2: Yes

2. Has the statistical analysis been performed appropriately and rigorously?

Reviewer #1: Yes

Reviewer #2: I don't know

3. Have the authors made all data underlying the findings in their manuscript fully available (please refer to the Data Availability Statement at the start of the manuscript PDF file)?

Reviewer #1: Yes

Reviewer #2: Yes

4. Is the manuscript presented in an intelligible fashion and written in standard English?

Reviewer #1: Yes

Reviewer #2: Yes

Reviewer #1: This manuscript describes an analysis of TB patient cost surveys using WHOs publicly available Health Equity Assessment Tool.

The paper is clearly written and highlights the disparity in catastrophic costs between individuals with drug-susceptible and drug-resistant TB. It also highlights differences between countries.

I do not have any major comments on the paper.

I have one minor question about the survey data. While these were carried out using a standardized WHO tool they were done in different years (2015-2020). Were there any changes to the WHO tool over this period? Or could differences in time period explain some of the between country differences observed e.g. changes in DR-TB regimens used?

Reviewer #2: This manuscript presents a cross-country analysis of catastrophic costs faced by tuberculosis-affected households across 19 low- and middle-income countries, using national TB Patient Cost Surveys and the WHO Health Equity Assessment Toolkit. The authors focus on inequalities by drug resistance status, comparing drug-susceptible TB and drug-resistant TB using multiple inequality measures (Difference, Ratio, Population Attributable Risk, and Population Attributable Fraction). The study highlights substantial financial burden and marked disparities, with drug-resistant TB–affected families consistently experiencing higher catastrophic costs.

Overall Assessment

Overall, this is a timely and policy-relevant manuscript that provides a valuable multi-country assessment of inequalities in catastrophic TB-related costs, with a particular focus on differences between DS-TB and DR-TB-affected households. The analysis is generally well conducted and clearly presented, and the use of multiple absolute and relative inequality measures strengthens its contribution to the TB equity literature. However, several issues related to conceptual framing, methodological transparency (particularly regarding the use of WHO HEAT), and interpretation of inequality measures would benefit from clarification to improve rigor, reproducibility, and policy relevance prior to publication.

Furthermore, in my opinion the overall analysis remains somewhat descriptive and would benefit from greater analytical depth. In its current form, the study largely relies on pre-loaded data and automated inequality measures within the WHO HEAT software, with limited examination/questioning of the assumptions behind the analysis, how reliable the results are, or what factors (in country eg TB/healthcare financing & drug availability, programmatic) might explain the differences seen. The study would also benefit from a more thorough explanation/justification for inequality measures and what other measures might be available (for benchmarking), or at least their limitations.

Line 20: Abstract introduction: “majority” of cases

Introduction

• The burden of TB is well described. The early paragraphs (1 and2) could be streamlined to reduce repetition regarding LMIC burden, poverty, and economic impact.

• The manuscript states that cross-country comparative evidence disaggregated by drug resistance status is limited. This claim could be strengthened by stating more explicitly from the outset what evidence already exists/more precise (policy-relevant) evidence gaps (e.g. lack of standardized inequality measures) and what this study might add, better linking the opening paragraph to the points that follow in the introduction. For example: “Previous multi-country studies have reported aggregate catastrophic cost estimates but have not applied standardized inequality measures or systematically disaggregated results by drug resistance status, limiting their usefulness for policy prioritization”.

• The definition of catastrophic costs (≥20% of annual household income) is clear, but perhaps the authors could be more explicit about the conceptual link between catastrophic costs and health inequity (eg we know drug resistance status is clinically relevant, but why is it an equity concern?)

• It would be helpful for the manuscript to state why HEAT is preferable, and also any potential limitations.

• Objective are well-formulated, but it may be helpful to clarify the distinction between comparing the magnitude of catastrophic costs (objective 2) and formally measuring inequalities (objective 3). Conceptually, they could potentially overlap.

Methods

• Please clarify whether the authors have used pre-loaded HEAT data or HEAT Plus (to include data on DR-TB and DS-TB)

• The surveys span 2015–2020, but the methods do not explain how differences in survey year were handled. This raises questions about cross-country comparability and should be acknowledged and discussed. It is noted that the discussion section mentions the health systems context and programmatic differences across countries have the potential to affect results and that further research is needed (lines 367-72).

• While D, R, PAR, and PAF are defined, it would strengthen the manuscript to briefly justify why these four measures were chosen over others, and how they complement each other in this specific TB context.

• Suggest including in the methods (or in a supplementary file) a description of how each inequality measure is calculated, as well as how the setting average is calculated.

• The study uses pre-loaded TB survey data and automated calculations within the WHO HEAT software. This means the authors rely on data and inequality measures already processed by WHO and calculated automatically by the software. It is unclear which analytical choices were made by the authors (for example, inclusion of countries, handling of subgroups) versus what HEAT did by default. Clarifying this would improve transparency, reproducibility, and reader confidence in the results.

• The authors note that D and R are unweighted and do not account for subgroup population sizes. It would be helpful to acknowledge this as a limitation and clarify whether it could bias the results, or if the DR-TB and DS-TB groups are similar in size, making unweighted measures appropriate.

• The HEAT software is described, but the exact version, date of access, and any parameter settings used for analysis are not reported. Providing these details improves reproducibility (perhaps as a supplement?).

• Results provide CI values, but there is no discussion in the methods of how uncertainty is handled/sensitivity analysis

• Sensitivity analysis would be helpful also for key inequality measures, eg, varying default assumptions

• Line 185: by contextualizing (not contextualize)

Results

Overall, results are comprehensive, clearly presented, and appropriately linked to the study objectives.

• The separation of catastrophic costs from inequality measures makes it easy for the reader to follow

• The inclusion of both country-specific percentages and confidence intervals provides transparency about the precision of estimates. Reporting both absolute (D, PAR) and relative (R, PAF) inequality measures strengthens the interpretation. This also allows the reader to appreciate both the magnitude and the potential impact of reducing inequalities, which is an important contribution to global TB policy discussions.

• The results clearly demonstrate that DR-TB-affected families generally face higher catastrophic costs than DS-TB families, with Burkina Faso as an informative exception. This directly addresses the study aim of assessing disparities.

• The text highlights countries with the highest disparities (e.g., Lesotho, Kenya, Papua New Guinea) and provides context for unusual findings (e.g., Burkina Faso), helping the reader understand where interventions may be most needed.

Suggestions / Clarification:

• The manuscript reports negative values for PAR and PAF and interprets them as indicating potential reduction in catastrophic costs if inequalities are addressed. Because the HEAT software reports PAR and PAF as negative for adverse indicators (where higher values are worse outcomes), it would strengthen the results if the authors explicitly explain that negative values are expected for adverse indicators under the WHO HEAT convention and clarify how these negative values should be interpreted in the context of their study.

• Table 2: please clarify in caption what the numbers represent (it is not clear in text, as sometimes results described as %, sometimes as magnitude. Eg some ratios (R) are described as “10.4 times reduction” in the text, which seems inconsistent with the table where PAF values are negative percentages. There may be a mismatch between narrative and table metrics that needs clarification. Also please check the -4 value for D in Burkina Faso.

• While CI ranges are provided for catastrophic cost estimates, no CIs are reported for D, R, PAR, or PAF. Including these would allow readers to assess statistical uncertainty in the inequality measures. See comments above in methods-sensitivity analysis, including varying assumptions, should also be described.

• In figure 1, the caption “Financial inequalities in catastrophic cost incurred by TB-affected families in Lesotho compared to 18 other low- and middle-income countries” could be clearer. It is unclear whether the figure compares all countries or highlights Lesotho specifically. Explicit mention of what the figure shows (absolute vs relative differences, etc.) would improve clarity.

• While disparities are reported, the results could briefly mention potential factors contributing to the observed differences (e.g., DR-TB treatment duration, additional non-medical costs), even if explored more in the discussion. This would help readers connect numbers to real-world implications.

Discussion

• The Discussion effectively summarizes key results, including the prevalence of catastrophic costs, differences by DR-TB vs DS-TB, and cross-country disparities. The use of absolute (D, PAR) and relative (R, PAF) measures is appropriately highlighted.

• The authors reference prior multi-country and single-country studies to situate their findings, demonstrating that the high burden of catastrophic costs among DR-TB households is consistent with global evidence.

• The Discussion clearly connects findings to actionable implications for TB control policy, UHC, and financial protection interventions (e.g., cash transfers, transport vouchers, decentralization of care).

• The authors note the cross-sectional design, reliance on self-reported data, country selection, programmatic differences, and the evolving landscape of DR-TB treatment. This transparency strengthens the credibility of the study.

• The suggestions for longitudinal studies, qualitative research, and intersectional analyses (gender, age, HIV status, rural/urban) are appropriate and forward-looking.

Further limitations/suggestions

• Lines 257-260: for clarity, I would suggest rephrasing the sentence as follows: “Notably, the PAR and PAF values (reported as negative due to HEAT conventions for adverse indicators) indicate that if the financial burden among DR-TB-affected families were reduced to the level of DS-TB-affected families, the overall prevalence of catastrophic costs could decline considerably.” Also this point seems to be repeated lines 283-5.

• It seems the text sometimes uses “times” reduction for PAF, which is inconsistent with HEAT output in percentage form. For example, Lesotho PAF = –10.4% does not mean costs would be reduced 10.4-fold. This should be corrected throughout the Discussion.

• The discussion mentions cross-sectional design and unmeasured confounding, but it could be more explicit that observed disparities cannot be interpreted as causal effects of DR-TB on catastrophic costs, only as associations.

• The anomaly in Burkina Faso (DR-TB households slightly less affected than DS-TB) is noted, but the Discussion could briefly explore why PAR/PAF are negative there and how this affects interpretation, linking back to HEAT conventions.

• The paper appropriately highlights disparities between drug-susceptible (DS-TB) and drug-resistant TB (DR-TB) households, which is central to the study’s objectives. However, it does not explore additional sub-groups such as age, sex, socioeconomic status, or urban/rural residence. Considering these factors could provide a more nuanced understanding of financial inequalities and identify other vulnerable populations that may benefit from targeted interventions. Though reference to unmeasured confounders lines 280-82, I would suggest a brief discussion or acknowledgment of these potential sub-groups would to strengthen results and ultimately the implications for policy (or perhaps implications for further studies?).

Other (minor) suggestions

• Repetition of statements about DR-TB being a “critical axis of inequality” could be condensed.

• Some numerical examples (e.g., D and R in multiple countries) could be summarized more concisely to improve readability.

Overall conclusion and recommendations:

Overall, this manuscript provides a timely cross-country overview of inequalities in catastrophic TB-related costs across 19 LMICs and clearly shows that DR-TB-affected households face a greater financial burden. The use of multiple inequality measures and the inclusion of cross-country comparisons are strengths and make the findings relevant for global TB and financial protection discussions.

That said, a few issues should be addressed to improve clarity and strengthen the analysis. In particular, the interpretation of negative PAR and PAF values would benefit from clearer explanation, as these reflect WHO HEAT conventions for adverse indicators and could easily be misinterpreted by readers. While the focus on drug-resistance status is appropriate, it would also be helpful to acknowledge and discuss the potential role of other unmeasured sub-groups (such as age, sex, socioeconomic status, or urban–rural residence) in shaping the observed inequalities. In addition, some form of sensitivity analysis—exploring alternative assumptions within HEAT, such as reference groups, weighting, or catastrophic cost thresholds—would increase confidence in the robustness of the results.

More generally, the paper would be strengthened by going beyond the default HEAT outputs and engaging more explicitly with analytical choices, assumptions, and the underlying cost components driving the observed patterns. Even within the limits of the pre-loaded data, this added depth would improve transparency and enhance the paper’s overall contribution.

**Do you want your identity to be public for this peer review?** For information about this choice, including consent withdrawal, please see our Privacy Policy .

Reviewer #1: No

Reviewer #2: No

---

## [Decision Letter · Decision Letter 1]

10 Mar 2026

INEQUALITIES IN THE FINANCIAL BURDEN OF TUBERCULOSIS AMONG AFFECTED FAMILIES: EVIDENCE FROM 19 LOW- AND MIDDLE-INCOME COUNTRIES

PGPH-D-25-02325R1

Dear Mr Sesay,

We are pleased to inform you that your manuscript 'INEQUALITIES IN THE FINANCIAL BURDEN OF TUBERCULOSIS AMONG AFFECTED FAMILIES: EVIDENCE FROM 19 LOW- AND MIDDLE-INCOME COUNTRIES' has been provisionally accepted for publication in PLOS Global Public Health.

Best regards,

Alice Zwerling, PhD

Academic Editor

Reviewer Comments (if any, and for reference):

Reviewer's Responses to Questions

**Comments to the Author**

Reviewer #1: All comments have been addressed

Reviewer #2: All comments have been addressed

publication criteria ? Is the manuscript technically sound, and do the data support the conclusions? The manuscript must describe methodologically and ethically rigorous research with conclusions that are appropriately drawn based on the data presented.

Reviewer #1: Yes

Reviewer #2: Yes

3. Has the statistical analysis been performed appropriately and rigorously?

Reviewer #1: Yes

Reviewer #2: I don't know

4. Have the authors made all data underlying the findings in their manuscript fully available (please refer to the Data Availability Statement at the start of the manuscript PDF file)?

Reviewer #1: Yes

Reviewer #2: Yes

5. Is the manuscript presented in an intelligible fashion and written in standard English?

Reviewer #1: Yes

Reviewer #2: Yes

Reviewer #1: (No Response)

Reviewer #2: Thank you for addressing my comments.

**Do you want your identity to be public for this peer review?** For information about this choice, including consent withdrawal, please see our Privacy Policy .

Reviewer #1: No

Reviewer #2: No
